# From Nuclear Receptor Regulation to Spleen Activating and Accumulation Resolving Therapy: A Review of Traditional Chinese Medicine Against Diabetes and Inflammation

**DOI:** 10.3390/ijms26136345

**Published:** 2025-06-30

**Authors:** Jiawen Huang, Like Xu, Weiru Liu, Chuanquan Lin, Ying Tang, Chuangpeng Shen, Yong Gao

**Affiliations:** State Key Laboratory of Traditional Chinese Medicine Syndrome, Science and Technology Innovation Center, Guangzhou University of Chinese Medicine, Guangzhou 510405, China; huangjw0630@163.com (J.H.); xulike2024@163.com (L.X.); liuwr917@163.com (W.L.); chuanquanlin@gzucm.edu.cn (C.L.); tangying@gzucm.edu.cn (Y.T.)

**Keywords:** diabetes, nuclear receptors, metabolism, inflammation, drug, Chinese medicine

## Abstract

Nuclear receptors are proteins located in the nucleus that are involved in gene transcription and play an important role in regulating metabolism and inflammation. Systemic metabolic abnormalities and chronic inflammation in diabetic patients are associated with gene expression and activity of bile acid metabolism, lipid and carbohydrate metabolism, energy expenditure, and inflammation regulated by nuclear receptors. As a major metabolic organ, the nuclear receptor regulation signal of the liver is the key to regulating the dialog between the liver and other organs. In this review, we discuss the newly discovered role of hepatic nuclear receptor signaling in diabetes metabolism and inflammation and focus on recent advances in drug research targeting nuclear receptors in diabetes, including the use of traditional Chinese medicine.

## 1. Introduction

As a global chronic metabolic disease, the incidence of diabetes is increasing yearly, which brings a heavy burden to the health and social economy of patients. Nowadays, diabetes has emerged as a major healthcare challenge, with the number of people living with diabetes expected to exceed 1.31 billion by 2050, continuing to pose a severe test for public health systems. Multiple studies have shown that the prevalence of diabetes continues to rise globally [1]. According to the International Diabetes Federation 2021 (IDF), which released the 10th edition of the global diabetes map data, in the past ten years (2011–2021) the number of diabetes patients in China increased by 50 million and has now reached 140 million. This is expected to grow to 164 million by 2030. Metabolic disorders and inflammatory reactions play a key role in the occurrence and development of diabetes, and they influence and promote each other, leading to the deterioration of diabetes. As an important transcription factor, nuclear receptors are widely involved in metabolic regulation, inflammatory response, cell proliferation, and differentiation, and play an important role in the pathogenesis of diabetes mellitus and its complications.

In recent years, more and more studies have been devoted to exploring the role of nuclear receptors in diabetes, and new nuclear receptor targets or ligands have been continuously discovered, which may have an improving effect on diabetes, providing a new perspective for the in-depth understanding of the pathophysiological mechanism of diabetes, and also provide hope for the development of new therapeutic strategies. For example, the activation or inhibition of certain nuclear receptor targets may affect insulin secretion, action, and glycolipid metabolism processes, thereby regulating blood sugar levels. Research on nuclear receptor targets or ligands is expected to open up new avenues for diabetes treatment. In the field of traditional Chinese medicine, the relationship between spleen deficiency and diabetes is also a concern. Spleen deficiency is considered to be one of the important pathogenesis of diabetes, and the treatment of diabetes with spleen-strengthening drugs has become an important method of traditional Chinese medicine. Studies have shown that spleen-strengthening drugs can play a role in lowering blood sugar levels by regulating metabolic function and enhancing insulin sensitivity.

When conducting this review, we searched for literature in PubMed and other online databases. The keywords used included ‘diabetes’, ‘nuclear receptors’ (including nuclear receptors such as LXR, FXR, and PPARs), ‘metabolism’, ‘inflammation’, ‘drug’, ‘Chinese medicine’, and their combinations. Initially, 489 relevant articles were retrieved from PubMed, and 36 were retrieved from other online libraries (such as CNKI, ResearchGate, etc.). After screening, 305 articles that were not directly related to the topic were excluded, such as those whose research focus was not on the relationship between nuclear receptors and diabetes metabolism and inflammation, or those that studied the treatment of diabetes with Chinese medicine but did not involve the mechanism of nuclear receptors. The screening criteria mainly focused on high-quality studies whose research content closely revolved around the role of nuclear receptors in diabetes, the development of drugs targeting nuclear receptors, and the association between the therapeutic effects of Chinese medicine on diabetes and nuclear receptors. Finally, 148 articles were included in this review.

To sum up, it is of great significance to improve the prognosis and quality of life of diabetic patients to further study the mechanism of nuclear receptor targets or ligands in metabolic disorders and inflammatory response of diabetes, explore its improvement effect on diabetes, develop drugs targeting nuclear receptors, and further study the mechanism and efficacy of spleen deficiency drugs in the treatment of diabetes. This review will comprehensively summarize and analyze the research progress in these aspects, and provide new ideas and methods for the treatment of diabetes.

## 2. Nuclear Receptors and Metabolic Disorders in Diabetes

A core feature of diabetes mellitus (DM) is the disorder of glucose and lipid metabolism, mainly due to the lack of insulin or insufficient insulin action leading to the disorder of carbohydrate metabolism, resulting in hyperglycemia and urine sugar. The main manifestations are reduced glycogen synthesis and enhanced gluconeogenesis. Disorders of glucose metabolism are also involved in insulin resistance and impaired function of islet beta cells. In type 2 diabetes, the function of islet beta cells to secrete insulin is impaired, while the responsiveness of the target organ to insulin is reduced, leading to hyperglycemia. This insulin resistance not only affects glucose metabolism but also leads to abnormal fatty acid metabolism, further exacerbating metabolic disorders. Abnormal lipid metabolism is also an important part of metabolic disorders. An abnormal lipid metabolism is closely associated with diabetes-related inflammation and complications. Changes in lipid metabolism affects a variety of mechanisms such as changes in metabolic pathways, regulation of cell function, and accumulation of metabolites, with nuclear receptors playing a key role. Farnesoid X receptor (FXR), liver X receptor (LXR), peroxisome proliferator-activated receptor α (PPARα), retinoid acid receptor-related orphan receptor γ (RORγ), and pregnane X receptor (PXR) are of great significance to diabetes (Figure 1 and Table 1). This part focuses on the effects of FXR, LXR and PPARα on the glucose and lipid metabolism disorders in DM.

### 2.1. Multifaceted Roles of FXR in Regulating Glucose and Lipid Metabolism Disorders in Diabetes: Focusing on Bile Acids, Glucose, and Lipid Regulation

FXR belongs to the nuclear receptor superfamily and has a unique molecular structure and functional properties. It forms heterodimers, alone or with the retinoid X receptor (RXR), that specifically bind to FXR response elements on DNA, thereby regulating the transcription of numerous target genes [2]. This ability of transcriptional regulation promotes numerous linkages to glucose and lipid metabolism and plays an irreplaceable role in maintaining the homeostasis of glucose and lipid metabolism. In animal experiments, FXR gene knockout or overexpression significantly affects the blood glucose level, insulin function, and glucose metabolism process of mice [3], which fully demonstrates the key position of FXR in glucose and lipid metabolism, and makes it a research hotspot in the treatment of disorders of glucose and lipid metabolism.

#### 2.1.1. Regulation of Bile Acid Metabolism

Bile acid (BA), as an endogenous ligand of FXR, plays an important signaling role in glycolipid metabolism [4]. In the liver, FXR, upon activation, regulates key enzymes for bile acid synthesis, such as cholesterol 7α-hydroxylase (CYP7A1). On the one hand, FXR induces the expression of small heterodimer chaperone (SHP), and SHP interacts with liver receptor homologous-1 (LRH-1) to inhibit CYP7A1 transcription [5]. On the other hand, FXR reduces the expression of CYP8B1 in the liver by regulating the oncogene homolog (MAFG) of V-Maf muscular aponeurotic fibrosarcoma [6]. These two mechanisms work together to maintain normal circulation and homeostasis of bile acids in the liver and small intestine. The regulation of the bile acid metabolic balance not only affects its metabolism but also is closely related to other links of glucose and lipid metabolism, which lays a foundation for FXR to further regulate glucose and lipid metabolism.

#### 2.1.2. Regulation of Glucose Metabolism

In terms of insulin regulation in pancreatic beta cells, FXR activation promotes the increase in insulin release and enhances the cell’s resistance to lipid toxicity, which is crucial for maintaining the normal function of pancreatic beta cells and insulin secretion [7]. At the same time, FXR enhances insulin signaling by regulating insulin receptor substrate (IRS) proteins, affecting their phosphorylation state or interacting with downstream signaling molecules, and improving tissue sensitivity to insulin [8]. These effects promote glucose uptake and utilization and maintain blood sugar stability.

Intestinal FXR activation can promote glucagon-like peptide-1 (GLP-1) secretion [9]. As an important incretin, GLP-1 can not only directly promote the secretion of insulin by pancreatic beta cells, but also inhibit the secretion of glucagon. Glucagon increases blood sugar, and inhibition of its secretion by GLP-1 helps prevent the excessive increase in blood sugar [10]. Through this dual regulatory mechanism, gut FXR is indirectly involved in glucose homeostasis regulation.

In terms of glycogen synthesis and gluconeogenesis regulation, FXR agonists can increase the expression of glycogen synthase (GS) through phosphorylation of glycogen synthase kinase 3β (GSK-3β). FXR agonists regulate relevant signaling pathways to deactivate GSK-3β phosphorylation, release glycogen synthase activity, promote glycogen synthesis, and increase liver glycogen storage [11]. Increased glycogen storage in the liver helps maintain stable blood sugar when blood sugar is low while improving insulin resistance.

In terms of gluconeogenesis, FXR regulates the expression of genes such as SHP, phosphoenolpyruvate carboxykinase (PEPCK), and glucose-6-phosphatase (G6Pase), which are key gluconeogenic enzymes whose activity affects the rate of gluconeogenesis. FXR also promotes the transcription of genes such as Cyp7a1, glucocorticoid receptor (GR), and HNF-4, which indirectly affects gluconeogenesis [12]. Studies have shown that berberine and the Chinese herbal compound Yitangkang affect the expression of PEPCK and G6Pase to regulate blood sugar by activating FXR, FXR agonists increase the expression of glucose transporters (GLUTs), promote glucose uptake, and improve insulin resistance.

#### 2.1.3. Regulation of Lipid Metabolism

In terms of lipid metabolism, FXR regulates the synthesis, oxidation, and transport of lipids by modulating the expression of a series of genes. The activation of FXR can inhibit the synthesis of triglycerides in the liver, accelerate the reverse cholesterol transport, and reduce the accumulation of lipids in the liver. FXR regulates the expression of fatty acid oxidation-related genes, such as carnitine/organic cation transporter 2 (OCTN2), promotes fatty acid β-oxidation, and reduces liver lipid accumulation [8]. In addition, FXR inhibits the expression of sterol regulatory element binding protein and other genes, reduces lipid synthesis, and maintains the balance of liver lipid metabolism [13]. In adipose tissue, FXR regulates the activity of transcription factors such as PPARγ, influencing adipocyte differentiation and lipid storage, thereby maintaining whole-body lipid metabolism balance [14].

#### 2.1.4. Differences in the Effects of FXR on Type 2 Diabetes Mellitus (T1DM) and Type 2 Diabetes Mellitus (T2DM)

T1DM is caused by the autoimmune destruction of pancreatic β-cells, resulting in an absolute insufficiency of insulin secretion. Although FXR can promote insulin release and enhance the lipid-toxicity resistance of β-cells, due to the massive damage of β-cells, its effect on improving overall blood glucose control is limited. In relevant animal models, the increased insulin release after activating FXR cannot fully correct severe hyperglycemia.

T2DM is characterized by insulin resistance and relatively insufficient insulin secretion. In the liver, FXR inhibits gluconeogenesis by regulating the genes of key enzymes, reducing glucose production. It also enhances insulin signaling, improving the sensitivity of tissues to insulin. In the intestine, FXR promotes the secretion of GLP-1, which indirectly regulates blood glucose. Activating FXR can lower blood glucose levels and improve insulin resistance in T2DM patients and animal models.

T1DM is often accompanied by abnormal lipid metabolism. FXR regulates the expression of genes related to fatty acid oxidation and lipid synthesis in the liver. Additionally, it inhibits the synthesis of triglycerides in the liver, accelerates the reverse transport of cholesterol, reduces lipid accumulation, improves lipid metabolism disorders, alleviates the lipotoxic damage to pancreatic β-cells and other tissues, and is indirectly beneficial for blood glucose control.

In T2DM, lipid metabolism disorders are significant, and the regulatory role of FXR is crucial for disease control. Activating FXR inhibits fatty acid synthesis in the liver, promotes β-oxidation, reduces lipid accumulation in the liver and adipose tissue, improves insulin resistance, and reduces the risk of cardiovascular diseases. In T2DM animal models, after giving FXR agonists, the lipid content decreases, and insulin sensitivity increases.

### 2.2. Dual Roles of LXR in Glucose and Lipid Metabolism in T2DM: Promoting Lipid Synthesis and Regulating Glucose

LXR is a class of nuclear receptors with unique molecular structure and functional characteristics. It can respond to the activation of oxidized cholesterol derivatives, bind to retinoid X receptors to form heterodimers, regulates the expression of target genes, plays a key role in the reverse transport of cholesterol and lipid metabolism, and also have an important impact on glucose metabolism, participating in the regulation of the pathological process of T2DM [15].

#### 2.2.1. Regulation of Lipid Metabolism

LXR activation directly promotes the transcription of sterol regulatory element binding protein-1c (SREGBP-1C) gene, and after its mRNA translation and processing activation, it enters the nucleus to activate fatty acid synthase genes, such as acetyl-CoA carboxylase (ACC) and fatty acid synthase (FAS), etc., to accelerate fatty acid synthesis [16]. At the same time, LXR activation promotes the expression of the fatty acid binding protein (FABP) gene, increases the intracellular FABP content, which promotes the transport and storage of newly synthesized fatty acids, avoiding excessive accumulation at the synthetic site, and indirectly supporting lipid production [17]. In addition, LXR up-regulates the expression of key enzyme genes for triglyceride synthesis, such as GPAT, and may also affect the gene expression of other lipid synthesis pathway-related enzymes, such as phospholipid synthesis, and comprehensively promote lipid production [18]. LXR also acts synergistically with PPARγ to enhance the regulation of genes related to lipid metabolism, form a more stable transcriptional activation complex, promote gene transcription and expression, and significantly stimulate lipid production [19]. Although LXR activation can regulate lipid metabolism and inflammatory mediators gene expression, and play an anti-atherosclerosis role, it also stimulates lipid overproduction. Excessive accumulation of lipids in liver cells leads to hepatic steatosis and elevated triglyceride levels in the blood, which may lead to other metabolic or cardiovascular diseases and pose a challenge to body health [20].

#### 2.2.2. Regulation of Glucose Metabolism

LXR inhibits the liver gluconeogenesis process, reduces the expression of key enzymes such as PEPCK and G6Pase, weakens the liver’s ability to synthesize glucose from non-sugar substances, reduces the amount of glucose output to the blood, and helps to maintain the stability of blood sugar [21]. The regulation of LXR on the expression of these enzymes may be achieved through direct or indirect mechanisms. Glucose uptake and insulin signaling improve LXR agonist-induced GLUT4 expression, where GLUT4 mainly exists in peripheral tissue cells, and its increased expression promotes glucose transport to intracellular metabolic utilization and reduces blood glucose concentration. At the same time, LXR inhibits the activation of insulin resistance-related c-Jun amino-terminal kinase (JNK) signaling pathway, improves insulin signal transduction, restores insulin sensitivity of cells, and promotes glucose uptake and utilization [22]. LXR agonist T0901317 has been shown to improve insulin sensitivity and reduce blood sugar levels in animal models [23], but its application is limited because it may cause fatty liver and other adverse reactions, and further research is needed to reduce adverse reactions.

#### 2.2.3. Differences in the Effects of LXR on T1DM and T2DM

LXR’s role in regulating blood glucose in T1DM is indirect. It mainly reduces the damage of lipotoxicity to pancreatic β-cells by regulating lipid metabolism protects the function of the remaining β-cells, and indirectly affects blood glucose. However, since the main problem in T1DM is the massive destruction of pancreatic β-cells, LXR’s direct regulatory effect on blood glucose is not obvious. In animal models, activating LXR has a weak effect on improving blood glucose.

LXR plays a more direct role in regulating blood glucose in T2DM. It inhibits hepatic gluconeogenesis, reduces glucose output by decreasing the expression of key enzymes, induces the expression of GLUT4 to promote glucose transport and utilization, and inhibits the JNK signaling pathway to improve insulin signal transduction and the sensitivity of cells to insulin. In animal models, giving LXR agonists can lower blood glucose and increase insulin sensitivity.

T1DM patients have abnormal lipid metabolism. LXR can promote the reverse transport of cholesterol, regulate fatty acid synthesis and metabolism, and reduce lipotoxic damage. However, limited by the pathological characteristics of T1DM, its effect on improving lipid metabolism is restricted.

Lipid metabolism disorder is an important feature of T2DM, and LXR plays a key role in it. Activating LXR promotes fatty acid synthesis, while also promoting its transportation and storage. It also synergizes with PPARγ to regulate lipid metabolism. However, excessive activation of LXR may lead to hepatic steatosis.

### 2.3. Multifaceted Roles of PPARα in Metabolic Regulation in Diabetes: Lipids, Glucose, and Beyond

Peroxisome proliferation-activated receptors (PPARs) are ligand-activated nuclear receptor superfamilies consisting of PPARα, PPARγ, and PPARβ/δ. Each PPAR has a specific expression pattern in adipose tissue, liver, skeletal muscle, and heart. PPARα forms a heterodimer with the retinoid AX receptor (RXR) and binds to DNA to exert transcriptional regulation [24]. PPARα is abundant in highly active metabolic tissues such as the liver, heart, muscle, kidney, brown adipose tissue, and vascular wall cells [25,26,27]. This distribution enables PPARα to participate in the regulation of various metabolic processes in many important metabolic organs and tissues and is closely related to the disorder of glucose and lipid metabolism. Studies have shown that PPARα inhibits the uptake and utilization of glucose by promoting the uptake and utilization of fatty acids, thereby regulating the metabolic pathways of fatty acids and glucose in cells.

#### 2.3.1. Regulation of Lipid Metabolism

In the regulation of serum lipid levels, PPARα regulates lipid and lipoprotein metabolism and increases the expression of lipoprotein lipase (LPL) and apolipoprotein (apo) A-V [27]. LPL catalyzes the hydrolysis of triglycerides in chylomicron and very low-density lipoprotein and reduces serum TG levels [28]. ApoA-V affects the metabolic process of lipoprotein [29,30]. At the same time, PPARα up-regulates the expression of genes involved in the β-oxidation pathway, enhances the oxidative decomposition of fatty acids, and maintains the balance of fatty acid levels in the liver. In lipoprotein metabolism, PPARα changes the interaction between very low-density lipoprotein (VLDL) and CETP and reduces the level of atherogenic lipoproteins, which is of great significance for cardiovascular health [31,32]. PPARα activation increases the expression of apoA-I and A-II and enhances HDL synthesis. Furthermore, apoA-I activates lecithin cholesterol acyltransferase, promotes cholesterol esterification, and contributes to HDL maturation and function [33]. The increased activity of LPL provides a phospholipid source for HDL particles, while PPARα also increases the expression of ABCA1, ABCG1, and type I Class B scavenger receptor (SR-BI), promotes cholesterol effluents from macrophages and selective uptake of cholesterol ester by the liver, accelerates the reverse cholesterol transport, and reduces the accumulation of cholesterol in the body and the risk of cardiovascular diseases [34].

#### 2.3.2. Regulation of Glucose Metabolism and Other Aspects

PPARα inhibits gluconogenesis by increasing the expression of pyruvate dehydrogenase kinase 4 (PDK4), thereby reducing glucose production in the liver. This mechanism helps to lower blood sugar levels, especially during fasting or starvation states [35]. In the fasting state, PPARα enhances glucose intake, glycolysis, and glycogenolysis, while increasing fat oxidation and glucose-induced insulin secretion in islet beta cells, thereby improving glucose management and insulin sensitivity. In addition, PPARα is also involved in hunger perception, and its ligand oleyl-ethanolamine (OEA) reduces food intake or delays eating by activating PPARα, leading to weight loss. PPARαKO mice (PPARα knockout mice) exhibit severe hypoglycemia and hyperinsulinemia after fasting, and in age-related hyperglycemia, PPARαKO mice have reduced expression of certain genes involved in gluconogenesis and glycogen metabolism [35]. In T2DM, PPARα expression may be influenced by gene polymorphism, transcriptional regulation, and epigenetic mechanisms. Pparalpha agonists such as fenbutalate and Wy14643 can improve glucose homeostasis by enhancing insulin sensitivity in adipose tissue and muscle while protecting islet beta cell function [36]. Activation of PPARα is also associated with the browning of brown adipose tissue, promotes systemic lipid oxidation and glucose metabolism, and has a positive effect on energy expenditure and insulin sensitivity in obese mice [36]. In high-fat diet-induced and hereditary insulin resistance mouse models, PPARα-selective fibrate activation can reduce insulin resistance, increase the expression of enzymes involved in lipid oxidation, improve the intracellular environment, restore the function of insulin signal transduction pathways, improve cell sensitivity to insulin, promote glucose uptake and utilization, and reduce blood sugar levels [37]. In a mouse model of Alstrom syndrome, the activation of PPARα by the agonist WY14643 had a similar effect, reducing steatosis and improving insulin sensitivity [30]. PPARα activation may be involved in glucose homeostasis, and although it is known to indirectly affect blood glucose by regulating lipid metabolism, there may be other unrevealed regulatory pathways [32]. In terms of thrombosis, PPARα may affect platelet function and expression of clotting factors, and inhibit thrombosis, which is an important risk factor for cardiovascular diseases. In the improvement of vascular function, PPARα may regulate the function of vascular endothelial cells, promote the release of vasodilator factors, inhibit the effect of vasodilator factors, and improve vascular endothelium-dependent relaxation function [38]. These potential effects need to be further studied to further understand the mechanism of action of PPARα in T2DM and related complications, and to provide a basis for the development of therapeutic strategies.

The mechanisms by which PPARα, LXR, and FXR regulate blood glucose levels differ significantly. PPARα primarily improves insulin sensitivity by promoting fatty acid oxidation and reducing hepatic glucose output, with its main effects on the liver and muscles. Its activation can lower plasma-free fatty acid levels, alleviate lipotoxicity, and thereby indirectly improve insulin resistance. LXR, on the other hand, influences blood glucose by regulating cholesterol metabolism and potentially suppressing gluconeogenesis. Its activation promotes reverse cholesterol transport and inhibits key gluconeogenic enzymes (such as PEPCK and G6Pase). However, it may exacerbate hepatic lipid accumulation and insulin resistance by activating SREBP-1c and promoting lipogenesis, with its effects concentrated in the liver and adipose tissue. Meanwhile, FXR lowers blood glucose through multiple pathways, including suppressing gluconeogenesis, enhancing insulin signaling, and regulating gut hormones (such as GLP-1). Its activation inhibits FoxO1-mediated gluconeogenesis and promotes GLP-1 secretion through intestinal FXR, enhancing insulin release and suppressing glucagon, primarily acting in the liver and intestines. Overall, PPARα focuses on improving energy metabolism and lipid homeostasis, LXR exhibits metabolic contradictions (improving cholesterol metabolism but potentially worsening fatty liver), and FXR demonstrates greater potential in regulating blood glucose through multi-target synergy and improving metabolic disorders.

Nuclear receptors such as FXR, LXR, and PPARα (and PPARγ) play a complex and important role in the disorder of diabetic glucose and lipid metabolism. Through their unique molecular mechanisms, they interact synergistically in bile acid metabolism, insulin function, gluconeogenesis, lipid synthesis and transport, fatty acid oxidation, cholesterol reverse transport, inflammation, and other aspects. An in-depth study of the functions and regulatory mechanisms of these nuclear receptors will help to reveal the pathogenesis of T2DM, providing an important theoretical basis for the development of new treatment methods and drugs. Future studies need to further explore the interaction network between nuclear receptors, as well as their specific mechanisms of action under different cell types and physiological and pathological conditions, to achieve more accurate and effective diabetes treatment, improve the disorder of glucose and lipid metabolism in patients, and prevent and treat related complications. At the same time, we should also pay attention to the potential adverse reactions that may be caused by nuclear receptor activation, find a way to balance the therapeutic effects and safety, and bring more clinical benefits to diabetes patients. In addition, with the continuous deepening of research, it is expected to find more regulatory targets related to nuclear receptors, providing more possibilities for the development of new drugs, and promoting the continuous development of the field of diabetes treatment.

**Figure 1 ijms-26-06345-f001:**
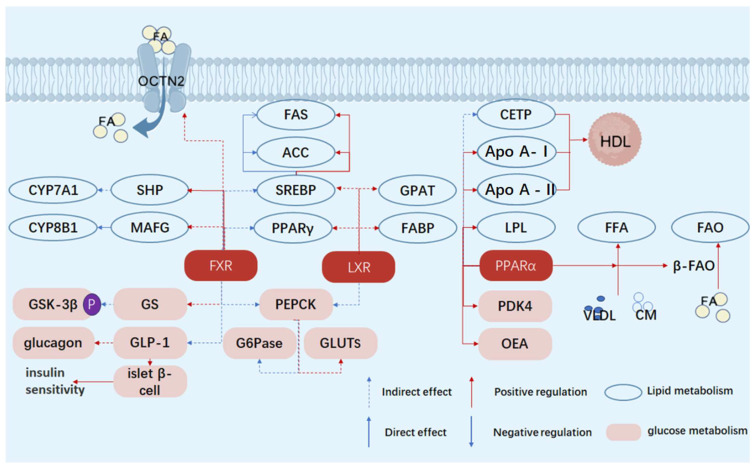
Schematic diagram of the mechanisms by which FXR, LXR, and PPARα regulate glucose and lipid metabolism. FA, fatty acid; FFA, free fatty acid; HDL, high-density lipoprotein; CETP, cholesteryl ester transfer protein; GPAT, glycerol-3-phosphate acyltransferase; apo A-I, apolipoprotein A-I; apo A-II, apolipoprotein A–II; VLDL, very low-density lipoprotein; LXR, liver X receptor; PPARα, peroxisome proliferator-activated receptor α; PPARγ, peroxisome proliferator-activated receptor γ; FXR, farnesoid X receptor; FAS, fatty acid synthase; ACC, acetyl-CoA carboxylase; FABP, fatty acid-binding protein; LPL, lipoprotein lipase; CM, chylomicron; SHP, small heterodimer partner; MAFG, MafG transcription factor; CYP7A1, cytochrome P450 7A1; CYP8B1, cytochrome P450 8B1; SREBP, sterol regulatory element-binding protein, OCTN2, organic cation transporter novel 2.

**Table 1 ijms-26-06345-t001:** Quick-view of the table of differences in metabolic and blood glucose regulatory functions of nuclear receptors PPARα, LXR, and FXR.

Characteristics	Receptors	Functions/Effects	References
Core metabolic pathways	PPARα	Fatty acid oxidation and energy metabolism	[39]
	LXR	Cholesterol metabolism and lipogenesis	[40]
	FXR	Bile acid metabolism and gluconeogenesis inhibition	[41]
Key points of blood glucose regulation	PPARα	Reduce hepatic glucose output and improve insulin sensitivity	[42]
	LXR	Cholesterol balance and potential inhibition of gluconeogenesis	[43]
	FXR	Directly inhibit hepatic glucose output and enhance GLP-1	[44]
Impact on the liver	PPARα	Reduce lipotoxicity and improve insulin sensitivity	[45]
	LXR	Promote lipogenesis (potential insulin resistance)	[16]
	FXR	Inhibit gluconeogenesis and improve insulin signaling	[46]
Tissue specificity	PPARα	Liver muscle	[47]
	LXR	Liver adipose tissue	[48]
	FXR	Liver and intestine	[49]
Drug applications	PPARα	Fibrates (such as fenofibrate)	[50]
	LXR	Limited (due to the risk of fatty liver)	[51]
	FXR	Obeticholic acid (in clinical trials)	[52]

## 3. Nuclear Receptors and Diabetes Inflammation

There is a complex interaction mechanism between diabetes and inflammation, and the two influence each other to form a vicious circle and aggravate the development of the disease (Figure 2 and Table 2).

### 3.1. Inflammation in Diabetes

Long-term hyperglycemia can cause non-enzymatic glycosylation of a variety of proteins in the body and form advanced glycosylation end products (AGEs). AGEs can bind to receptors on the cell surface and activate a series of signaling pathways, such as the nuclear factor-κB (NF-κB) pathway, and promote the expression and release of inflammatory factors such as tumor necrosis factor-α (TNF-α) and interleukin-6 (IL-6), thus triggering an inflammatory response [53].

In a state of insulin resistance, the insulin signaling pathway is impaired. On the one hand, the sensitivity of fat cells and liver cells to insulin is reduced, resulting in abnormal blood sugar metabolism. On the other hand, insulin resistance can cause lipid metabolism disorders in adipocytes, releasing excessive free fatty acids (FFAs). FFAs can activate inflammation-related signaling pathways such as NF-κB, induce the production of inflammatory factors, and trigger inflammation [54].

Oxidative stress increases the imbalance between oxidation and antioxidants in diabetic patients, and hyperglycemia, hyperlipids, and other factors can promote an increase in reactive oxygen species (ROS) production. ROS can damage intracellular biomacromolecules such as DNA, proteins lipids, etc.; thus, activating intracellular stress signaling pathways, including those related to inflammation, promoting the secretion of inflammatory cytokines and triggering inflammation [55].

### 3.2. Inflammation Exacerbates Diabetes

#### 3.2.1. Islet Beta Cell Damage

Inflammatory factors can directly act on islet beta cells, induce β cell apoptosis, and reduce insulin secretion by activating NF-κB and other signaling pathways. For example, TNF-α can interfere with the expression of genes related to insulin synthesis and secretion in beta cells, affecting the normal production of insulin, and thus, aggravating diabetes. Worsening insulin resistance in an inflammatory state, results in inflammatory factors interfering with insulin signaling pathways, and further reducing the sensitivity of cells to insulin. For example, IL-6 can inhibit the tyrosine phosphorylation of insulin receptor substrates (IRS) by activating signal transduction proteins and transcriptional activators (STAT) and other pathways, blocking the transmission of insulin signals to the downstream, making insulin resistance more serious and blood sugar difficult to control.

#### 3.2.2. Influence on Blood Glucose Metabolism

Inflammation can affect the uptake, utilization, and storage of glucose by the liver, muscle, and other tissues. For example, inflammation can inhibit the activity of key glucose-metabolizing enzymes such as glucokinase in the liver, which reduces glucose uptake and glycogen synthesis in the liver and promotes gluconeogenesis, which increases blood sugar and exacerbates diabetes.

### 3.3. LXR’s Regulation of Inflammatory Response

LXR is a class of nuclear receptors, including two subtypes LXRα and LXrβ. LXRα is mainly highly expressed in tissues involved in lipid metabolism, while LXRβ is widely expressed in a variety of tissues. In the immune system, macrophages, dendritic cells, and neutrophils express two subtypes, and B lymphocytes mainly express LXRα. The T lymphocyte population can express LXRβ or both. The liver X receptor (LXR) forms a heterodimer with the retinoid X receptor (RXR) and binds to LXREs to regulate the expression of target genes. LXR is involved in the regulation of the expression of multiple genes related to lipid metabolism. The increase in cholesterol levels can activate LXREs, promoting cholesterol efflux and participation in cholesterol metabolism, making LXR a sensor for cholesterol homeostasis. Additionally, the LXR also has anti-inflammatory effects [20].

LXR’s inhibition of inflammation can be realized through the regulation of lipids. A large number of studies have shown that cholesterol metabolism is directly related to the occurrence of inflammation. The increase in low-density lipoprotein (LDL) levels in the body can cause cholesterol accumulation, promote endothelial dysfunction and activation, and cause the expression of pro-inflammatory cytokines. Inhibiting the efflux of cholesterol, and reducing the level of cholesterol accumulation can effectively inhibit the inflammatory response. LXRα activates regulated cholesterol homeostasis and inhibits inflammatory cytokine production and inflammatory signaling pathways. In macrophages, LXR can inhibit the expression of a variety of pro-inflammatory genes, and LXRα is the main transcriptional regulator of ABCA1 and ABCG1 in macrophages, which limits the accumulation of cholesterol in macrophages and other peripheral cells mainly by increasing the expression of ABCA1 and ABCG1 [56]. In an inflammatory response, LXR can regulate the expression of a series of inflammation-related genes, such as interleukin-6 (IL-6), IL1β, etc. These genes are usually regulated by transcription factors such as NF-κB, and the activation of LXR can interfere with signaling pathways such as NF-κB, thereby inhibiting the transcription of inflammatory genes. The specific mechanism involves conformational changes after ligand binding, such as SUMOylation modification. In macrophages, SUMO LXR interacts with the actin-binding protein CORONIN2A (CORO2A) to prevent actin recruitment to inflammatory gene promoters, thereby inhibiting transcription and inhibiting the production of inflammatory factors by macrophages [57,58]. In the LPS-induced inflammation model, the activation of LXR reduces the production of inflammatory factors, thereby alleviating the inflammatory response [59]. For example, cyanidin-3-o-β-glucoside (C3G) inhibits the LPS-mediated NF-κB signaling pathway by activating LXRα-ABCG1-dependent cholesterol efflux [58]. LXRs also have a regulatory effect on the NLRP3 inflammasome. Studies have shown that LXR agonist treatment restricts the activation of the NLRP3 inflammasome, while the restriction effect of the agonist is ineffective after LXRα/β knockout [60].

### 3.4. ER’s Regulation of Inflammatory Response

The estrogen receptor (ER) that regulates the inflammatory response can be located in the cell membrane, cytoplasm, or nucleus. The classical nuclear receptor is located in the nucleus. The ER mediates a variety of signal pathways, such as MAPK/ERK signal transduction pathway, PI3K/Akt signal transduction pathway, JNK signal transduction pathway, etc.

Research shows that ER interacts with nuclear factor-κB (NF-κB). MJ Evans et al. gave oophorectomy C57BL/6 mice an atherosclerotic diet and found that it induced the expression of multiple inflammatory genes in the liver. Mice treated with 17α-acetylide estradiol (EE) inhibited the expression of atherosclerotic diet-induced inflammatory genes and had no regulatory effect on the expression of cholesterol metabolism genes. EE treatment did not affect nuclear translocation or DNA binding of NF-κB, suggesting that estrogen receptors work by interfering with the transcriptional activity of NF-κB. Mice treated with ER antagonist ICI could block the inhibitory effect of EE on inflammatory genes, proving that ER mediates the inhibitory activity of EE [61].

### 3.5. FXR’s Regulation of Inflammatory Response

FXRs are expressed in the liver and intestine, and can regulate the expression of target genes by binding to ligands to form heterodimers, and can also indirectly inhibit gene transcription. In the course of inflammation, the FXR receptor can play a role through a variety of pathways, in monocytes, macrophages, and dendritic cells, FXR activation can reduce the production of cytokines, and play an anti-inflammatory role. For example, in macrophages, the activation of FXRs inhibits the release of inflammation-related cytokines; thus, the activation of FXRs plays a crucial role in regulating inflammatory responses by inhibiting the production of pro-inflammatory cytokines, thereby reducing the severity of inflammation. In the intestine, the activation of FXRs can inhibit the expression of genes related to inflammation, thereby alleviating intestinal inflammation. For example, treating mice with FXR agonist OCA can reduce the expression of pro-inflammatory cytokine genes such as IL-1β and IL-6 in the colon. Meanwhile, FXR deficiency impairs intestinal barrier function, making the liver more susceptible to inflammatory signals from bacterial sources, further demonstrating the importance of FXRs in maintaining intestinal inflammatory homeostasis [62].

In the course of sepsis, the increase in the bile acid level is positively correlated with the mortality of the disease. As a damage-associated molecular model (DAMPs), bile acids induce a long-term influx of calcium ions and work collaboratively with ATP to activate signal 1 and signal 2 of the NLRP3 inflammasome in macrophages. Studies have shown that FXR negatively regulates the NLRP3 inflammasome by interacting with NLRP3 and Caspase-1. Compared with FXR-deficient mice, the level of pro-inflammatory cytokine IL-1β in FXR-overexpressed mice was decreased, indicating that FXR can inhibit the production of pro-inflammatory cytokines by negatively regulating the NLRP3 inflammation, thereby regulating the inflammatory response [63].

In addition, FXRs indirectly influence inflammatory responses by regulating processes such as lipid synthesis and lipoprotein secretion in the liver. By regulating lipid metabolism, lipid balance can be maintained and inflammation caused by lipid metabolism disorder avoided. Lipid metabolism disorder can cause oxidative stress and endoplasmic reticulum stress, and then activate inflammatory signaling pathways, such as NF-κB, resulting in the production of a large number of inflammatory factors. FXR regulates lipid metabolism to help stabilize the intracellular environment and reduce the activation of inflammatory signals. FXRs reduce the synthesis and secretion of triglyceride-rich lipoproteins in the liver, reduce plasma triglyceride levels, and improve lipid metabolism disorders, thereby reducing inflammatory responses. FXR reduces plasma triglyceride levels by inhibiting the expression of liver transcription factor sterol regulatory element-binding protein-1c (SREBP-1c) and its lipogenesis target gene. SREBP-1c plays a key role in lipid synthesis, and its inhibition reduces fat synthesis in the liver. Activation of FXR can induce the expression of PPARα, which can promote the oxidation of fatty acid β and accelerate the catabolism of fatty acid, thereby reducing the accumulation of fat in the liver. FXRs can further regulate inflammatory responses by regulating cholesterol metabolism and affecting inflammatory signaling pathways [64].

### 3.6. PPARs Regulating Inflammatory Response

PPARs have been proven to play a role in a variety of chronic diseases, such as diabetes, cancer, coronary atherosclerosis, etc. [65,66]. Of the three subtypes of PPARs, PPARγ has been studied the most.

The function of PPARγ is associated with a variety of inflammatory responses and autoimmune diseases. Under different microenvironments, macrophages will be polarized into pro-inflammatory M1 macrophages or anti-inflammatory M2 macrophages to adapt to environmental changes [67,68]. In the progression of liver inflammatory diseases such as nonalcoholic fatty liver disease, PPARγ plays a role by mediating Kupffer cells, resident macrophages of the liver. In high-fat diet-induced hepatic steatosis, Kupffer cells polarize toward M1, producing pro-inflammatory cytokines such as Interleukin-1 (IL-1), Interleukin-6 (IL-6), tumor necrosis factor α (TumorNecrosisFactor-α, TNF-α), causing inflammation. Various studies have shown that PPARγ can promote the polarization of Kupffer cells towards M2, counteracting the pro-inflammatory effect of M1-type macrophages, and playing a role in improving inflammation and reducing cell damage [68,69].

PPARα is also involved in the regulation of inflammatory responses. Various inflammatory parameters of PPARα-deficient mouse peritoneal macrophages were significantly increased under LPS/IFN-γ culture [70]. PPARα inhibits NF-κB activation in LPS-induced synovial fibroblasts [71]. In addition, PPARα has a sex-specific role in the development of T-cell responses and T-cell-mediated autoimmune diseases. In male mice, PPARα is highly expressed, and its loss leads to an enhanced Th1 response and a diminished Th2 response, which is associated with enhanced NF-κB and c-Jun activities [72]. In lipid metabolism, PPARα also plays a crucial role. PPARα is extremely abundant in the liver and can participate in the regulation of lipid metabolism in the liver, including the decomposition, β-oxidation, synthesis, transportation, and storage of fatty acids [73,74]. When PPARα is activated by free fatty acids, intracellular β-oxidation is enhanced, and ATP and ketone body production are increased, so that PPARα can use free fatty acids to provide the energy required by the body in the state of nutritional deficiency [75]. Under starvation, the absence of PPARα in mouse liver leads to decreased β-oxidation activity of cell mitochondria, which is unable to fully utilize free fatty acids, resulting in liver lipid metabolism disorder and severe hepatic lipid accumulation [76]. When mice are fed a high-fat diet (HFD), liver PPARα deficiency can cause severe liver steatosis and hepatitis [77].

Members of the PPAR family are recognized regulators of lipid metabolism, mitochondrial biogenesis, and energy balance. Their activation has important effects on the function of oxidized tissues and organs such as cardiomyocytes, liver, and muscle. PPARs have been a potent therapeutic target for treating metabolic disorders for many years, making it an attractive topic for researchers. At present, the role of different members of this nuclear receptor family in the differentiation and function of immune cells is gradually being explored. It can be said that PPARs are established mediators of macrophage polarization and key determinants of T-cell activation, expansion, and differentiation [78].

**Figure 2 ijms-26-06345-f002:**
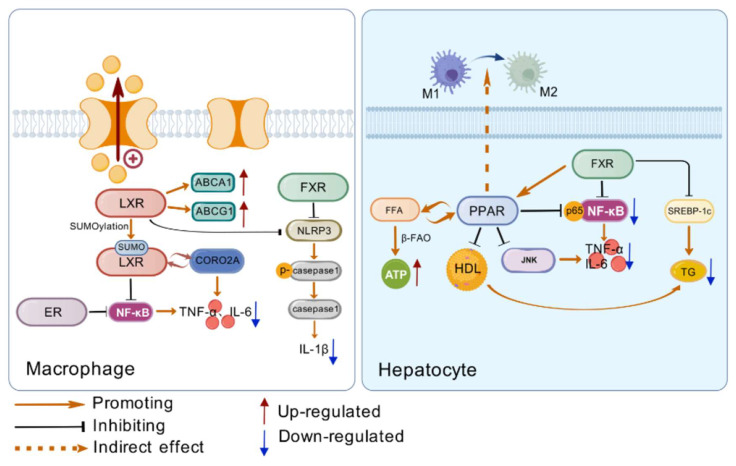
(**Left side**) The way LXR, ER, and FXR regulate inflammation in macrophages. LXR can reduce intracellular lipid accumulation by regulating the expression of lipid metabolism genes, and all three nuclear receptors can reduce the production of inflammatory factors by regulating inflammatory pathways. (**Right side**) Regulation of inflammatory response by PPAR and FXR in hepatocytes. PPAR can promote the polarization of Kupfer cells towards M1, accelerate the β-oxidation of free fatty acids, and inhibit the activation of NF-κB. FXR induced PPAR expression and inhibited the expression of lipid synthesis genes. Created with https://biogdp.com/.

**Table 2 ijms-26-06345-t002:** Comparison table of key information on inflammatory regulation by nuclear receptors LXR, ER, FXR, and PPARs.

Characteristics	Receptors	Cell Types/Effects	References
Regulatory Cells	LXR	Macrophages	[79]
	FXR	Monocytes, macrophages, dendritic cells, and intestinal cells	[80,81]
	PPARs	Kupffer cells (PPARγ)	[68,69]
Regulatory Mechanisms	LXR	Regulates lipid metabolism to inhibit inflammation; Interferes with NF-κB pathway through conformational changes	[82,83]
	ER	Interacts with NF-κB to inhibit its transcriptional activity	[84]
	FXR	Reduces pro-inflammatory cytokines; Regulates lipid metabolism to avoid inflammation	[85,86]
	PPARs	PPARγ promotes M2 polarization of Kupffer cells; PPARα inhibits NF-κB activation	[68,69,87]
Impact on Diabetic Inflammation	LXR	Alleviates inflammation and improves insulin resistance	[88]
	ER	Inhibits liver inflammation genes	[84]
	FXR	Reduces systemic inflammation and maintains intestinal inflammation homeostasis	[89]
	PPARs	Improves liver inflammation and regulates immune cell function	[90]

## 4. New Nuclear Receptor Targets for the Treatment of Diabetes

### 4.1. FXR Improves Glycolipid Metabolism by Regulating ZBTB18

ZBTB18 is a multi-zinc finger protein, belonging to the DNA-binding transcription factor, which is closely related to neuronal differentiation and development and can regulate cell differentiation and function by inhibiting the expression of specific genes [91,92].

In recent years, research on ZBTB18 has gradually increased, and its positive role in various diseases has been slowly discovered. In glioblastoma, ZBTB18 has been found to inhibit the SREBP gene by inhibiting the activity of the CTBP-LSD1 complex, ultimately reducing lipid synthesis and improving lipid storage [93]. The regulation of lipids by ZBTB18 can also be realized through FXR. Studies have found that ZBTB18 can directly bind to the promoter region of the FXR gene, thereby activating FXR transcription [94]. In the context of diabetes, the activation of FXR can improve metabolic status in various ways. In terms of lipid metabolism, FXR can stimulate its downstream target genes after activation and accelerate the β-oxidation (FAO) process of fatty acids in the liver, which helps to consume excess lipids and improve the imbalance between lipid synthesis and consumption [95]. After FXR is knocked out, the protective effect driven by ZBTB18 protein is weakened, and the stimulating effect of FAO induced by ZBTB18 protein overexpression is reduced [94].

In addition to lipid regulation, ZTB18 also acts on the body’s inflammatory response, which is again accomplished by FXR. ZBTB18 can inhibit the activation of NLRP3 inflammasome in macrophages through FXR-mediated expression of CLTC protein. In vivo and in vitro experiments revealed that the absence of ZBTB18 in the liver can aggravate the inflammatory response and increase the expression of NLRP3, ASC, Caspase-1, and other inflammasome-related proteins, while the overexpression of ZBTB18 can reverse these changes and inhibit the inflammatory response. In the liver fibrosis model induced by a methionine–choline-deficient (MCD) diet, overexpression of ZBTB18 can inhibit the expression of inflammatory genes by activating FXR and its downstream target genes, reducing the aggregation of inflammatory cells in the liver, and thus, reducing the accumulation of fibrosis-related proteins such as α-SMA and Col1a1 in the liver. This protects mice from liver fibrosis [94].

### 4.2. KLF16 Regulates PPARα to Improve Glycolipid Metabolism

KLF16 (liver Kruppel-like factor 16) is a transcription factor that plays an important role in regulating metabolism and disease processes. Recent studies have shown that the expression of KLF16 is related to the progression of bladder cancer and other cancers, and interference with its expression can inhibit the growth and migration of cancer cells [96].

In the context of lipid metabolism disorders, KLF16 plays a role mainly through the regulation of nuclear receptors. The study found that in hepatic steatosis, KLF16 expression decreased, while in KLF16 deficient environments, liver steatosis increased and insulin resistance increased, and KLF16 overexpression reduced liver steatosis in db/db and HFD mice, suggesting that KLF16 can play a role in lipid metabolism. KLF16 can directly bind to the promoter of PPARα, thereby activating PPARα transcription, accelerating fatty acid β oxidation, alleviating mitochondrial stress and oxidative stress, reducing lipid accumulation, and improving insulin resistance in db/db and HFD mice. In PPARα-deficient environments, the protective effect of KLF16 disappeared, demonstrating that KLF16 improves lipid deposition and insulin resistance in a PPARα-dependent manner [97].

In addition, KLF16 also has the effect of reducing intracellular oxidative stress. During the progression of diabetes, high levels of blood glucose can cause mitochondria to produce reactive oxygen species (ROS) [98]. The increase in ROS will not only damage mitochondrial function, but also damage pancreatic beta cells, affecting insulin secretion, causing insulin resistance, and forming a vicious cycle [99]. Studies have confirmed that KLF16 overexpression can significantly upregulate antioxidant-related genes such as Nrf2 and SOD2, thereby significantly reducing ROS produced in a high-fat environment. After KLF16 or PPARα is knocked out, antioxidant gene expression is inhibited and ROS production increases significantly, indicating that KLF16 alleviates oxidative stress and maintains intracellular redox balance in a PPARα-dependent manner [97]. This provides new treatment ideas for alleviating oxidative stress to improve diabetes.

### 4.3. Nur77 Affects AMPK Glucose Metabolism

AMPK is a key metabolic switch that controls key target proteins of metabolic pathways by phosphorylation, increases skeletal muscle fatty acid oxidation, and inhibits triglyceride synthesis and lipogenesis. AMPK also stimulates glucose uptake through muscle contraction through a PI3K-independent mechanism [100]. Studies have shown that the effect of metformin, a first-line diabetes drug, in inhibiting hepatic glucose production and improving hyperglycemia in patients is related to AMPK activation [91,101]. This shows that AMPK plays an important role in regulating glucose metabolism. Studies have found that Nur77 is involved in regulating the localization of LKB1 and the activation of AMPK [102]. Specifically, Nur77 binds to LKB1 and affects its stability in the nucleus, thereby affecting the transport of LKB1 to the cytoplasm and activating AMPK. Nur77 increases glucose transport in mouse skeletal muscle cells under lipotoxic conditions by activating p38MAPK [100]. Hence, Nur77 is not only directly involved in the regulation of glucose metabolism but also indirectly affects glucose metabolism through other signaling pathways. In β cells, the knockout of Nur77 leads to disorders in amino acid, lipid, and glucose metabolism [103]. This further confirms the important role of Nur77 in glucose metabolism.

In the field of diabetes treatment, traditional drugs have certain limitations, so it is urgent to explore new treatment strategies. Identifying new targets or new ligands for regulating nuclear receptors has become a focal point of current research. As an important class of transcription factors, nuclear receptors are widely involved in the regulation of diabetes-related metabolic pathways. By accurately identifying new nuclear receptor-related targets or ligands, it is possible to open up new research perspectives and entry points for in-depth intervention in the complex metabolic network of diabetes.

This direction holds great potential in the development of diabetes drugs. Compared with existing drugs, drugs developed based on new targets or new ligands are expected to break the efficacy bottleneck of traditional drugs and achieve targeted regulation of key links in the onset of diabetes with their more precise mechanisms of action. This type of drug may not only be more targeted in terms of therapeutic effect, and can directly attack the core crux of diabetic metabolic disorders, significantly improving the efficiency of blood sugar control, but also has greater advantages in terms of safety, and can effectively reduce the risk of adverse reactions. Therefore, this provides more reliable protection for long-term drug treatment.

## 5. Drugs Targeting Nuclear Receptors

In the complex pathological process of diabetes, nuclear receptors are key regulatory nodes, exhibiting extremely close and intricate connections with diabetes, traditional Chinese medicine (TCM), and inflammation. As critical regulators of intracellular gene transcription, nuclear receptors play a significant role in the onset and progression of diabetes. In terms of metabolic disorders, nuclear receptors such as FXR, LXR, and PPARα maintain the body’s energy balance by regulating bile acid, lipid, and carbohydrate metabolism. FXR modulates bile acid metabolism, thereby influencing lipid and glucose metabolism; LXR plays a pivotal role in lipid synthesis and glucose metabolism; PPARα is involved in fatty acid oxidation and glucose metabolism regulation. Dysfunction of these receptors can directly exacerbate metabolic disorders in diabetic patients.

Inflammation and diabetes interact with each other, forming a vicious cycle. Chronic hyperglycemia, insulin resistance, and oxidative stress trigger inflammatory responses, which in turn damage pancreatic β-cells, worsen insulin resistance, and disrupt glucose metabolism. Nuclear receptors also play an important role in inflammation regulation. For instance, LXR, FXR, ER, and PPARs can modulate the expression of inflammation-related genes, inhibit inflammatory signaling pathways, and reduce the production of inflammatory factors.

Notably, TCM demonstrates unique advantages in regulating nuclear receptor functions. Numerous TCM formulations and single herbs can act on nuclear receptors to regulate metabolic disorders and inflammatory states in diabetic patients. For example, Yunu Jian (Jade Woman Decoction) can clear stomach heat, nourish kidney yin, improve metabolic disorders, and alleviate inflammatory responses by modulating nuclear receptors. Active components in single herbs such as astragalus polysaccharides from *Astragalus* and allantoin from Chinese yams can regulate nuclear receptor-related signaling pathways, enhance insulin sensitivity, improve glucose and lipid metabolism, and simultaneously reduce inflammatory damage. This suggests that TCM may regulate nuclear receptors through multiple targets and pathways, thereby providing new strategies and directions for diabetes treatment and highlighting the immense potential of TCM in the comprehensive management of diabetes.

Several drugs are currently on the market that target nuclear receptors involved in the pathogenesis of diabetes. These drugs mainly involve PPARs.

### 5.1. PPARs Agonists

Rosiglitazone and pioglitazone are two marketed PPARγ agonists. Pioglitazone enhances insulin signaling and insulin sensitivity by activating PPARγ receptors [104]. Rosiglitazone also reduces blood glucose levels and improves islet beta cell function [105]. Pioglitazone has significant anti-inflammatory effects, reducing myocardial ischemia/reperfusion injury, inhibiting CYP11B2 expression and aldosterone production, and reducing CD40/CD40L expression. Rosiglitazone also reduces inflammatory states [104]. Pioglitazone improves endothelial dysfunction, reduces levels of hypersensitive CRP, reduces increases in prothrombotic and inflammatory cytokines, increases plasma adiponectin, and lowers blood pressure. Pioglitazone significantly reduces triglyceride, increases high-density lipoprotein cholesterol (HDL-C), and improves lipid metabolism [106]. PPARγ activators protect islet beta cells from fatty acid-induced damage, promote insulin secretion, and reduce insulin resistance. These drugs also improve the mitochondrial function of islet beta cells, reduce beta cell apoptosis, and prevent islet amyloid deposition [104]. Pioglitazone improves GLUT4 expression in adipose tissue, thereby enhancing insulin signaling [107]. Although rosiglitazone and pioglitazone have a positive effect on the treatment of T2DM, they may also cause side effects such as weight gain, edema, and osteoporosis. Therefore, the use of these drugs needs to be considered in combination with their efficacy and potential risks, as well as patient-specific conditions and compliance [107]. In summary, rosiglitazone and pioglitazone work together in T2DM through multiple mechanisms, including improved insulin sensitivity, anti-inflammatory, cardiovascular protection, improved lipid metabolism, and protection of islet beta cells.

Fenofibrate acts as a PPARα agonist, and by up-regulating FGF21, activates PI3K/Akt/GSK-3β/Fyn signaling pathway, up-regulates Nrf2 antioxidant function, thereby delaying the progression of diabetic nephropathy [108].

### 5.2. Metadichol

Metadichol is a novel nanolipid that treats diabetes by binding to the vitamin D receptor (VDR). Metadichol can bind to the VDR as an inverse agonist, and this binding may explain its effects on a range of biomarkers, including blood pressure, blood lipids, inflammatory markers, and vitamin C levels. The binding effect of Metadichol to the VDR is similar to that of the natural ligand 1,25(OH)_2_D of the VDR. This ligand can participate in the regulation of γ-glutamyl transpeptidase (γ-GT), an enzyme that can upregulate the level of glutathione (GSH). Therefore, it can be inferred that Metadichol may also have the ability to upregulate GSH. In addition, Metadichol can activate the antioxidant transcription factor NRF2 and simultaneously inhibit the activity of the pro-inflammatory transcription factor NF-κB, thereby promoting the expression of antioxidant-related genes, including GSH. GSH is a key substance in the antioxidant regeneration pathway and an important antioxidant in cells. Sufficient GSH can ensure the recycling of vitamin C, thus maintaining and increasing the vitamin C level in the body [109,110]. This regulation of the vitamin C level helps protect cells from oxidative damage, which may be of great significance to diabetic patients because there is often enhanced oxidative stress in the bodies of diabetic patients.

### 5.3. Retinoic Acid Receptor-Related Orphan Receptor Gamma (RORγ) and SR1001

RORγ plays an important regulatory role in immune and inflammatory responses. The role of RORγ in diabetes is mainly related to an immune response and inflammation. For example, RORγt is the main transcription factor of IL-17A, which exacerbates islet inflammation by directly inducing beta cell apoptosis and locally increasing cytokines and chemokines. SR1001 is a selective RORα/γ inverse agonist with significant effects on diabetes. In a study in non-obese diabetic mice, SR1001 significantly reduced the incidence of diabetes and islet inflammation, while reducing the expression of pro-inflammatory cytokines, particularly TH17-mediated cytokines, and increasing the frequency of regulatory T-cells (Tregs) [111]. SR1001 has also been found to inhibit Th17 cell function, thereby effectively alleviating autoimmune diseases and type 1 diabetes. Therefore, by inhibiting the activity of RORγt, SR1001 can reduce islet beta cell damage and hyperglycemia caused by diabetes. SR1001 showed significant therapeutic potential in diabetic mice by inhibiting RORγt activity and decreasing Th17 cell differentiation and function, suggesting that SR1001 may be an effective diabetic drug, especially in the treatment of autoimmune diabetes [112].

The development and application of these drugs demonstrate the importance of nuclear receptors in the treatment of diabetes, and also suggest the potential for drug development targeting more nuclear receptors in the future.

### 5.4. Nuclear Receptor Drugs in Development

At present, new nuclear receptor agonists for diabetes treatment are being actively developed.

SPPARMα (selective peroxisome proliferator-activated receptor-α modulator) is a novel drug with higher potency and selectivity compared to traditional non-selective PPARα agonists. For example, fenofibrate, a SPPARMα agonist, has shown significant reductions in triglycerides and increases in high-density lipoprotein cholesterol (HDL-C) in animal models, and is also excellent at inhibiting postprandial-based triglyceride increases [113]. There are significant differences in pharmacological action between SPPAMα and PPARγ agonists. SPPARMα mainly plays a role in regulating lipid metabolism, anti-inflammatory effect, and improving liver function, while PPARγ agonists are more involved in the regulation of lipid metabolism and cell signaling pathways. Therefore, the two may have different application prospects in the treatment of hyperlipidemia and other metabolic diseases [114]. Double agonists have also emerged in recent years, Tesaglitazar, as a PPARα/γ double agonist, has shown anti-inflammatory effects in animal models, which may be related to its effect on pro-inflammatory cells in adipose tissue [115].

FXR agonists such as GW4064 can improve tacrolimus-induced post-transplant diabetes and regulate the expression of related glucose metabolism genes [116]. In recent years, the research on FXR agonists has shifted from steroidal to non-steroidal compounds, and many structurally novel FXR agonists have been found through high-throughput screening. At present, a variety of FXR agonists have entered the clinical research stage worldwide, such as Obeticholic acid, EDP-305, Cilofexor, Tropifexor, LMB763, PXL007, etc. These drugs may show potential in the treatment of diabetes, but further clinical studies are needed to verify their efficacy and safety [117,118,119].

## 6. Traditional Chinese Medicine

Modern medicine mainly controls the course of diabetes by oral hypoglycemic drugs or subcutaneous insulin injection, but some patients’ blood glucose levels are difficult to control in an ideal manner, resulting in frequent complications. In contrast, traditional Chinese medicine has shown unique advantages in the treatment of diabetes and related complications [120]. In recent years, traditional Chinese medicine has made significant progress in the treatment of diabetes. In TCM, this disease belongs to the category of diabetes. Its core pathogenesis is “yin deficiency as the root and dryness-heat as the tip”. The disease affects the lungs, stomach, spleen, and kidneys. According to the specific symptoms, its pathological characteristics can be manifested as lung dryness, stomach heat, spleen deficiency, and liver and kidney deficiency. If the course of the disease is prolonged, it can lead to the deficiency of viscera complicated with various pathological changes. For example, deficiency of liver and kidney yin may cause deafness, blindness of eyes, deficiency of blood vessels may lead to numbness of limbs, and blood stasis and yin deficiency and dryness-heat may easily lead to furuncle and other diseases. The nature of the disease is mostly mixed with deficiency and excess, and the basic principle of treatment is “clearing heat and moistening dryness, nourishing yin and generating body fluid”. Clinically, diabetes patients often show symptoms such as thirst and polydipsia, polydipsia, fatigue, and emaciation, which can be divided into upper wasting-thirst, middle wasting-thirst, and lower wasting-thirst according to TCM syndrome differentiation. The treatment principle mainly revolves around “engendering fluids to quench thirst, nourishing yin and clearing heat”, and the symptomatic treatment has a remarkable curative effect. Chinese medicine treatment is mainly to nourish yin and moisten dryness, nourish yin, tonify kidney, and assist yang, and more Chinese medicines containing sweet and bitter herbs are selected to achieve a therapeutic effect.

### 6.1. Spleen Deficiency Compound

#### 6.1.1. Nourishing Yin, Moistening Dryness—Yunv Decoction

Yunu Decoction is derived from “Complete Works of Jingyue”. The gypsum in its prescription is pungent, sweet, and cold. It can clear yangming stomach heat, stimulate the production of fluids, and quench thirst, so it is a sovereign medicine. Prepared rehmannia root is used to nourish kidney water deficiency. The sovereign and minister match, clear stomach fire and strengthen kidney water, which can treat both deficiency and excess. Combined with sweet and bitter cold Rhizoma Anemarrhenae, it can help gypsum clear stomach heat and stop thirst and help prepare Rehmannia to nourish Shaoyin and strengthen kidney water. It has the meaning of mutual production of gold and water and also has the effect of promoting the production of fluids and moistening dryness. Achyranthes bidentata can tonify the liver and kidneys and guide blood downward, so it is used as an assistant and courier. The combination of all herbs can clear stomach heat, nourish kidney yin, combine sweet and cold methods, treat stomach and kidney simultaneously, purge excess, and tonify deficiency. It has been suggested for stomach heat yin deficiency syndrome of Shaoyin deficiency and Yangming excess. In clinical practice, patients with diabetes have the main symptoms of “drinking, polyphagia, polyuria and weight loss”. The syndrome belongs to excessive stomach heat, which is the same as the indication of Yunujian. This can improve clinical symptoms [121,122].

The clinical study observed the effects of Yunu decoction on blood glucose and insulin-β cell function in patients with T2DM with gastric incandescent syndrome. The clinical efficacy, gastric heat incandescent syndrome score, blood glucose index (fasting blood glucose FBG, 2h postprandial blood glucose P2hBG, HbA1c), islet beta cell function index (fasting insulin FINS, insulin resistance index HOMA-IR, islet beta) of the experimental group (Yunu decoction plus or minus + conventional treatment) and the control group (conventional treatment only) were compared. According to the changes in cell function index HOMA-β, the addition and reduction in Yunv decoction in the treatment of gastric heat incandescent T2DM is effective, which can effectively relieve clinical symptoms, reduce blood sugar levels and improve islet beta cell function [123]. In a study by Liang Ruifeng’s team, it was proved that achyranthes bidentate can improve islet beta cell function and insulin sensitivity by cooperating with other ingredients in Yunu Decoction, thus regulating blood sugar levels [124].

#### 6.1.2. Nourishing Yin and Tonifying Kidney—Liuwei Dihuang Pill

The Liuwei Dihuang pill comes from “Children’s medicine Syndrome Straight Formula”, treating kidney yin deficiency of diabetes. This recipe specifically uses cooked Rehmannia and is a royal medicine, nourishes Yin and fills the marrow. *Dioscorea* and Cornus are two herbal drugs that supplement the liver, spleen, and kidneys, three viscera, and can astringent essence, not only tonifying kidney and reinforce essence, but also tonifying spleen to help the source of producing qi and blood, and considered a minister medicine. These three drugs warm the spleen and kidney to treat kidney yin deficiency, kidney essence deficiency syndrome. During the method of supplementing, “turbid” must be emptied to preserve “clarity”, so that yin essence can be replenished. Therefore, alisma rhizome is used to remove dampness and purging turbid to prevent the greasy of cooked Rehmannia. Peony bark clears the deficiency fire, and inhibits the warm astringent of Cornus officinale meat, Poria and *Dioscorea* match, tonifying the spleen and helping spleen health.

In addition to the general symptoms of diabetes disease in clinical patients with this syndrome, it can be seen that “frequent urination and turbidness of urine are similar to the symptoms of fat cream, waist, and knee tenderness”, which is the same as the treatment of the Liuwei Dihuang Pill and has a positive curative effect [125]. Nian Junyu et al. [126] proved that the Liuwei Dihuang Pill combined with metformin could significantly enhance insulin sensitivity and promote the decomposition of sugars and lipids in the body, thus achieving the effect of lipid regulation and glucose lowering. The combination of the Liuwei Dihuang Pill and englaglitazine can inhibit the activity of glucokinase (TC), block the process of insulin exocytosis, increase the oxidative stress of islet beta cells, and reduce insulin secretion. At the same time, the metabolism of triglyceride (TG) can lead to ectopic fat deposition through the release of free fatty acids, increase liver glycogen synthesis, affect glucose metabolism and utilization, and reduce insulin clearance and sensitivity, thus achieving the purpose of hypoglycemia [127].

#### 6.1.3. Tonifying Kidney and Yang—Shenqi Wan

The Shenqi pill from “Golden Chamber Essentials”, tonifys the kidneys to assist Yang, degenerates kidney Qi, and treats kidney Yang deficiency syndrome. As reported, “when men quench their thirst, they urinate more, so that they drink a bucket and urinate a bucket, which is mastered by the kidney Qi pill” [128]. In the prescription, cooked Rehmannia is the sovereign medicine, nourishing Yin and filling marrow. In the book of Materia Medica, it is pointed out that dry Rehmannia is an essential medicine of tonifying kidneys and has the top quality of boosting Yin and blood. “Used cornus as the minister medicine, nourishing liver and kidney, tonifying kidney qi; *Dioscorea* invigorating spleen qi, consolidating kidney essence; Add cinnamon twig, aconite root, warm kidney to help Yang, tonify Yang qi and encourage kidney qi. With Tuckahoe, alisma, and tree peony bark to inhibit the deficiency of Yang floating”. The combination of various medicines is not a drastic tonic, but seeking Yang within Yin, generating Yang qi. It is clinically effective in the treatment of early diabetes [129].

Gao Yuwen et al. found that the Jinkui Shenqi Pill can regulate the Nrf2/HO-1 pathway, increase protein levels, reduce blood sugar in rats, and improve oxidative stress damage in rat kidney tissue [130]. Zhang Jingzu et al. [131] proved by using relevant statistical methods that the Jinkui Shenqi Pill significantly affected the treatment of yin-yang and yin-yang deficiency-type diabetic nephropathy and had fewer side effects than the Western medicine group. Zhao Junju et al. [132] have shown that diabetic patients with kidney-yang deficiency have amino acid metabolism disorders, and Shenqi Pills can effectively improve this metabolic abnormality. The Shenqi Pill can regulate the metabolic disorder of energy, fat, and amino acid in diabetic kidney-yang deficiency model rats, and then exert a hypoglycemic effect. Wu Xiaocui et al. [133] proved that the Jiawei Jinkui Shenqi Pill can reduce IL-6 activation of microglia and reduce the function of macrophages, thus reducing the level of IL-8 and other pro-inflammatory factors, reducing inflammatory response, and achieving therapeutic effect.

### 6.2. Example of Spleen Deficiency Single Flavor Medicine

Meng Xiao et al. [134] found that both the extract of *anemarrhena rhizome* and the extract of *Sanqi* showed good hypoglycemic effects, which could reduce the fasting blood glucose level and improve glucose tolerance in hyperglycemic model rats. At the same time, these extracts can also reduce the level of serum glycated serum protein (GSP), up-regulate insulin (INS) and high-density lipoprotein cholesterol (HDL-C), significantly reduce total cholesterol (TC), triglyceride (TG), and low-density lipoprotein cholesterol (LDL-C), showing the potential of regulating glucose and lipid metabolism. The hypoglycemic effect of quercetin is mainly through two mechanisms: first by increasing the glucose uptake of muscle cells, promoting liver glycogen synthesis and insulin secretion [135,136], and second, inhibiting α-glucosidase activity, thereby reducing the process of converting carbohydrates into glucose in the small intestine [137], effectively treating diabetes. According to Li Congyu et al. [138], sweet Chinese herbs have significant curative effects in the treatment of T2DM (T2DM), and its mechanisms include regulation of glucose and lipid metabolism, reduction in islet tissue damage, enhancement of insulin sensitivity, anti-oxidative stress, and inhibition of inflammation. By acting on the intestinal bitter receptor (TAS2R), bitter Chinese medicine activates phosphodiesterase (PDE) and phospholipase C (PLC) signaling pathways, and promotes the release of glucagon-like peptide-1 (GLP-1) from intestinal endocrine L cells. GLP-1 further effectively reduces postprandial blood glucose levels by stimulating insulin secretion and inhibiting glucagon release, showing good therapeutic potential [139].

#### 6.2.1. Qi-Tonifying Herb

*Astragalus* is a commonly used Qi-tonifying Chinese medicine, which has the effects of invigorating the spleen and qi, strengthening Yang and consolidating the exterior, inducing diuresis to alleviate edema, invigorating qi and promoting blood circulation, etc. These efficacy characteristics enable it to play a role in the treatment of diabetes and its complications. For example, *Astragalus* can improve diabetic nephropathy (DKD) due to its effect of dispersing swelling, strengthening the spleen, and raising Yang, and its extract can reduce proteinuria in DKD rat models and improve renal function [140]. The bioactive ingredients in *Astragalus* have also been shown to improve diabetes. Its main active components include Astragaloside and Astragalus polysaccharide. Astragaloside, for example, is a natural and effective free radical scavenger that can lower blood sugar levels, improve insulin resistance, and reduce oxidative stress [141]. In addition, Astragalus polysaccharide has been proven to improve insulin resistance and glucose metabolism through various signaling pathways, and it can also protect islet beta cells and improve insulin sensitivity [142].

*Dioscorea* is sweet, and flat, and has the effect of tonifying the spleen and stomach, engendering fluid and moistening lungs, tonifying kidneys, and consolidating astringent essence. Its main active ingredient is Allantoin. *Dioscorea* belongs to the kidney tonifying Chinese medicine and has a certain effect on improving diabetes and its complications. Studies have shown that yam gruel alone or combined with metformin can reduce blood glucose and TG levels in diabetic rats by activating the AMPK/ACC/CPT-1 pathway and improving the disorder of lipid metabolism in the liver, which provides a new idea for diabetic dietary therapy [143]. In addition, allantoin, the active ingredient in *Dioscorea* gruel alone, reduces the blood glucose level in diabetic rats in a dose-dependent manner. Studies have shown that allantoin can promote the release of β-endorphin in the isolated adrenal medulla of diabetic rats, thereby increasing the expression of muscle glucose transporter 4 (GLUT4) gene and enabling the body to increase glucose absorption. To achieve the effect of lowering blood sugar levels [144].

#### 6.2.2. Blood Tonic Herb

Angelica in Chinese medicine is considered to have the effect of tonifying blood and promoting blood circulation, regulating menstrual pain, and moistening the bowel, and belongs to the blood-tonifying Chinese medicinal materials. Its main active component is Angelica polysaccharide. Modern research has found that *Angelica sinensis* can play an active role in the treatment of diabetes. The active components of *Angelica sinensis* mainly include polysaccharides, coumarins, and so on. Among them, Angelica polysaccharide is one of the main active components. These polysaccharides improve metabolic disorders in people with diabetes by regulating the gut microbiota. For example, 60% ethanol extract of *Angelica sinensis* can enhance the intestinal barrier function of diabetic mice, regulate the imbalance of intestinal microbiota, increase the number of beneficial bacteria, and reduce the blood sugar level [145]. In addition, a polysaccharide (ASP-Ag-AP) isolated from the aboveground part of *Angelica sinensis* has been found to exert anti-inflammatory effects by inhibiting the TLR4/MyD88/NF-κB signaling pathway [146].

#### 6.2.3. Yin-Tonifying Herb

*Dendrobium* is a kind of traditional Chinese medicine commonly used for tonifying Yin. Its main active components are Dendrobium polysaccharides and alkaloids (such as shihunidine and shihunine). Recent studies have confirmed that *Dendrobium* contains a variety of active ingredients that can improve a variety of diseases. In the treatment of diabetes, Dendrobium polysaccharides have been found to promote hepatic glycogen formation and inhibit hepatic gluconeogenesis by affecting the glucagon-mediated cAMP-PKA and Akt/FoxO1 signaling pathways, thereby improving blood glucose levels. Dendrobium polysaccharide can also change the structure of liver glycogen, reverse its instability, and inhibit the degradation of liver glycogen [147]. In addition, the alkaloids in *Dendrobium* also have the effect of treating diseases. For example, shihunidine and shihunine isolated from *Dendrobium* have inhibitory effects on Na^+^/K^+^-ATPase in rat kidneys and have shown significant hypoglycemic effects in diabetic rats [148].

## 7. Conclusions

A key regulator of intracellular gene transcription is nuclear receptors that not only play an important role in the pathological process of diabetic metabolic disorders and chronic inflammation but also connect the signal communication network between the liver and other metabolic organs. In this paper, the role of PPAR, FXR, LXR, and other nuclear receptor families in diabetes metabolism regulation, inflammation control, and related pathologic mechanisms were systematically reviewed. At the same time, the research and development of drugs targeting nuclear receptors continues to deepen, from chemical synthetic drugs to traditional Chinese medicine compounds, providing a multidimensional perspective for the precision treatment of diabetes.

Compared with existing reviews, this review has several unique features. Firstly, it focuses on the connection between nuclear receptors and the traditional Chinese medicine (TCM) therapy of strengthening the spleen and eliminating stagnation. It delves into the mechanisms by which TCM regulates nuclear receptors. For instance, it explores how certain traditional Chinese medicine compounds interact with nuclear receptors to improve glucose and lipid metabolism and inflammation in diabetic patients. Specifically, this review elaborates on how the herbs in traditional Chinese medicine formulas such as Yunu Decoction, Liuwei Dihuang Pill, and Shenqi Pill may act on nuclear receptors to exert therapeutic effects. Secondly, it comprehensively analyzes the role of traditional Chinese medicine in regulating nuclear receptors for the treatment of diabetes. It not only lists the blood-glucose-lowering and blood-lipid-lowering effects of various traditional Chinese medicine herbs and compounds but also explores their potential mechanisms related to nuclear receptor regulation. Compared with many existing reviews, this represents a more in-depth and systematic analysis.

In diabetic metabolic disorders, the function of nuclear receptors regulating bile acid metabolism, lipid metabolism, and carbohydrate metabolism is gradually clear, and the disturbance of its signaling pathway directly affects the whole body’s energy balance. In inflammatory pathology, nuclear receptors not only regulate the expression of pro-inflammatory and anti-inflammatory factors but also are closely related to insulin resistance through oxidative stress and other pathways. In addition, other roles of nuclear receptors in diabetes, such as regulating cell proliferation, mitochondrial function, and gut microbiota balance, also provide new directions for future research.

In recent years, the development of new drugs based on nuclear receptor targets has made rapid progress, and some of them have entered the clinical trial stage. For example, the application of FXR agonists in bile acid metabolism is promising for PPAR family agonists to improve lipid metabolism disorders. At the same time, as a potential nuclear receptor modulator, traditional Chinese medicine shows its unique advantages by regulating multi-target signaling pathways. In particular, the efficacy of TCM compounds and monomer compounds (such as Scutellaria Scutellaria, Rehmannia, etc.) in improving metabolic disorders and inflammation in diabetic patients with spleen deficiency syndrome has become a hot research topic in the world.

In the future, research on nuclear receptors should focus more on the analysis of specific regulatory mechanisms and clarify their core role in the pathological network of diabetes. In addition, the interaction between nuclear receptors and other molecular signaling pathways should be further explored, providing new ideas for multi-target combination therapy. In the field of drug development, integrating modern drug design and traditional Chinese medicine resources and optimizing nuclear receptor target drugs through multi-omics technology will promote the development of diabetes precision medicine. Furthermore, network pharmacology and molecular docking methods hold great potential in revealing the action targets of traditional Chinese medicine components. Future research could apply these methods to deeply explore the interaction mechanisms between traditional Chinese medicine components and nuclear receptors, providing a theoretical basis for the development of more effective diabetes treatment regimens. In summary, nuclear receptor research not only deepens our understanding of the pathological mechanism of diabetes, but also lays an important foundation for the development of innovative therapies, and provides new hope for the all-round management of diabetes patients.

## Data Availability

No new data were created or analyzed in this study. Data sharing is not applicable to this article.

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
