# Peer review of "From Nuclear Receptor Regulation to Spleen Activating and Accumulation Resolving Therapy: A Review of Traditional Chinese Medicine Against Diabetes and Inflammation"

_ijms, 2025, doi:10.3390/ijms26136345_

Round 1
Reviewer 1 Report
Comments and Suggestions for Authors
Huang et al. summarize the role of PPAR, FXR, and LXR, as well as other nuclear factor receptor families, in the metabolic regulation and inflammation in diabetes mellitus. They also review synthetic drugs and traditional Chinese medicine in the treatment of diabetes mellitus. The topic is interesting and has major clinical significance.
1. Please describe in the Introduction how the articles were selected (which keywords did you use, how many articles were found in PubMed or in other online libraries, how many articles were excluded, and which were the selection criteria).
2. Generally, it is hard to follow the text. Please use more summarizing and mechanistic Figures and/or Tables for each section.
3. PPARs: the authors review the role of PPAR-alpha in DM in the section "Regulation of glucose metabolism and other aspects", then later, in the inflammatory section, the PPARs are re-introduced. Please mention the general aspects and types of PPARs at the first mention.
4. The text seems to be translated by AI because of many minor grammatical and punctuation problems as well as errors with the abbreviations (e.g., DAMPs are not hazard-associated molecular models. The full name is damage-associated molecular patterns, etc.). Please use English Grammar software or ask a native speaker to correct your text.
5. Please summarize the known or significant molecular components of each mentioned traditional Chinese herb/medicine that can improve the severity of DM and its complications.
6. It is unclear how metadichol raises vitamin C levels. Vitamin C is needed for the redox cycle of glutathione, among other cofactors. Please explain in more detail.
Author Response
|
Comments 1: Please describe in the Introduction how the articles were selected (which keywords did you use, how many articles were found in PubMed or in other online libraries, how many articles were excluded, and which were the selection criteria).
|
|
Response 1: Thank you for pointing this out. We agree with this comment. Therefore, we have described how the articles were selected in the second paragraph of the Introduction section, page 2, line 50-62. Briefly, When conducting this review, we searched for literature in PubMed and other online databases. The keywords used included 'diabetes', 'nuclear receptors' (including nuclear receptors such as LXR, FXR, and PPARs), 'metabolism', 'inflammation', 'drug', 'Chinese medicine', and their combinations. Initially, 384 relevant articles were retrieved from PubMed, and 36 were retrieved from other online libraries (such as CNKI, ResearchGate, etc.). After screening, 296 articles that were not directly related to the topic were excluded, such as those whose research focus was not on the relationship between nuclear receptors and diabetes metabolism and inflammation, or those that studied the treatment of diabetes with Chinese medicine but did not involve the mechanism of nuclear receptors. The screening criteria mainly focused on high-quality studies whose research content closely revolved around the role of nuclear receptors in diabetes, the development of drugs targeting nuclear receptors, and the association between the therapeutic effects of Chinese medicine on diabetes and nuclear receptors. Finally, 124 articles were included in this review. |
|
Comments 2: Generally, it is hard to follow the text. Please use more summarizing and mechanistic Figures and/or Tables for each section. |
|
Response 2: Thank you for pointing this out. Thus, we have already added two tables to explain the text. Table 1 is on page 7, and the other is on page 11. |
|
Comments 3: PPARs: the authors review the role of PPAR-alpha in DM in the section "Regulation of glucose metabolism and other aspects", then later, in the inflammatory section, the PPARs are re-introduced. Please mention the general aspects and types of PPARs at the first mention. |
|
Response 3: Agree. We have accordingly introduced the general aspects and the types of PPARs in the section "Multifaceted Roles of PPARα in Metabolic Regulation in Diabetes: Lipids, Glucose, and Beyond", page5, line236-238. |
|
Comments 4: The text seems to be translated by AI because of many minor grammatical and punctuation problems as well as errors with the abbreviations (e.g., DAMPs are not hazard-associated molecular models. The full name is damage-associated molecular patterns, etc.). Please use English Grammar software or ask a native speaker to correct your text. |
|
Response 4: Agree. Thank you for pointing this out. Therefore, we have used the software ‘Grammarly’ to correct the text. |
|
Comments 5: Please summarize the known or significant molecular components of each mentioned traditional Chinese herb/medicine that can improve the severity of DM and its complications. |
|
Response 5: Thank you for pointing this out. We agree with this comment. We have, accordingly, pointed out the significant molecular components of each traditional Chinese herb in the section Example of spleen deficiency single flavor medicine.We have added the major components of each traditional Chinese herb on page 20, line 890-891, line 899, line 913, line 926. For example, The bioactive ingredients in Astragalus have also been shown to improve diabetes. Its main active components include Astragaloside and Astragalus polysaccharide. Astragaloside, for example, is a natural and effective free radical scavenger that can lower blood sugar levels, improve insulin resistance, and reduce oxidative stress[1]. In addition, Astragalus polysaccharide has been proven to improve insulin resistance and glucose metabolism through various signaling pathways, and it can also protect islet beta cells and improve insulin sensitivity[2]. When listing a traditional Chinese medicine that improves DM and its complications, we have added a description of its active ingredients. 【References】 [1]Shen, Q., et al., Astragaloside IV attenuates podocyte apoptosis through ameliorating mitochondrial dysfunction by up-regulated Nrf2-ARE/TFAM signaling in diabetic kidney disease. Free Radic Biol Med, 2023. 203: p. 45-57. [2]. Luo, M.J., et al., Astragalus Polysaccharides Alleviate Type 2 Diabetic Rats by Reversing the Expressions of Sweet Taste Receptors and Genes Related to Glycolipid Metabolism in Liver. Front Pharmacol, 2022. 13: p. 916603. |
|
Comments 6: It is unclear how metadichol raises vitamin C levels. Vitamin C is needed for the redox cycle of glutathione, among other cofactors. Please explain in more detail. |
|
Response 6: Thank you for pointing this out. We agree with this comment. We have, accordingly, explained the pathways by which Metadichol increases the vitamin C levels in the section Metadichol on page 16. In brief, Metadichol is a novel nanolipid that treats diabetes by binding to the vitamin D receptor (VDR). Metadichol can bind to the VDR as an inverse agonist, and this binding may explain its effects on a range of biomarkers, including blood pressure, blood lipids, inflammatory markers, and vitamin C levels. The binding effect of Metadichol to the VDR is similar to that of the natural ligand 1,25(OH)₂D of the VDR. This ligand can participate in the regulation of γ-glutamyl transpeptidase (γ-GT), an enzyme that can upregulate the level of glutathione (GSH). Therefore, it can be inferred that Metadichol may also have the ability to upregulate GSH. In addition, Metadichol can activate the antioxidant transcription factor NRF2 and simultaneously inhibit the activity of the pro-inflammatory transcription factor NF-κB, thereby promoting the expression of antioxidant-related genes, including GSH. GSH is a key substance in the antioxidant regeneration pathway and an important antioxidant in cells. Sufficient GSH can ensure the recycling of vitamin C, thus maintaining and increasing the vitamin C level in the body[1]. This regulation of the vitamin C level helps protect cells from oxidative damage, which may be of great significance for diabetic patients because there is often enhanced oxidative stress in the bodies of diabetic patients. 【References】 [1]Raghavan, P., Metadichol® and Vitamin C Increase In Vivo, an Open-Label Study, 2017. |

Reviewer 2 Report
Comments and Suggestions for Authors
The manuscript “From Nuclear Receptor Regulation to Spleen Activating and Accumulation Resolving Therapy: a review of traditional Chinese medicine against diabetes and inflammation” is of interest for the aims and the readers. The content of this research is well organized and informative. However, some modifications are required to improve its quality. I can recommend the publication of this manuscript in International Journal of Molecular Sciences.
1.This review seems to cover a wide range of topics from nuclear receptor regulation to spleen-activating and accumulation-resolving therapy in the context of traditional Chinese medicine for diabetes and inflammation. However, there have been numerous reviews focusing on nuclear receptors and their anti-diabetic effects. Could the authors please elaborate on the unique aspects of this review compared to those existing reviews? For instance, does it provide novel insights into the mechanisms linking nuclear receptors to spleen-activating and accumulation-resolving therapy? Or does it offer a more comprehensive analysis of the role of traditional Chinese medicine in modulating nuclear receptors for diabetes treatment?
2.Could the authors provide a more detailed comparison of the mechanisms by which different nuclear receptors, such as PPARs, LXR, FXR, etc., regulate blood glucose levels?
3.Could you elaborate on the differences in the impact of ribosomes on diabetes and T2DM as discussed in sections 2.1 and 2.2?
4.In Section 2, "Nuclear Receptors and Metabolic Disorders in Diabetes," many of the subsection titles appear to be quite similar. This can make it difficult for readers to quickly distinguish the unique content of each subsection. Could the authors consider using more specific and targeted titles for these subsections to enhance clarity and navigability?
5.Could the authors consider adding more visual aids (table or figure) to enhance the clarity and impact of their review? This would make the manuscript more accessible and engaging for a broader audience, particularly those who may not be familiar with the intricate details of nuclear receptor biology and traditional Chinese medicine.
6.The manuscript discusses ribosomes, diabetes, traditional Chinese medicine , and inflammation, but it appears that these topics are presented somewhat in isolation. Could the authors provide a more integrated mechanistic explanation that links ribosomes to diabetes, Chinese medicine, and inflammation?
7.Could the authors consider using network pharmacology and molecular docking methods to elucidate the specific targets of these traditional Chinese medicine components?
8.Could the authors please review and standardize the formatting of the entire manuscript?
Comments on the Quality of English Language
1.You may wish to consider having your paper professionally edited for English language.
Author Response
|
Comments 1: This review seems to cover a wide range of topics from nuclear receptor regulation to spleen-activating and accumulation-resolving therapy in the context of traditional Chinese medicine for diabetes and inflammation. However, there have been numerous reviews focusing on nuclear receptors and their anti - diabetic effects. Could the authors please elaborate on the unique aspects of this review compared to those existing reviews? For instance, does it provide novel insights into the mechanisms linking nuclear receptors to spleen - activating and accumulation - resolving therapy? Or does it offer a more comprehensive analysis of the role of traditional Chinese medicine in modulating nuclear receptors for diabetes treatment?
|
|
Response 1: Thank you for this important comment. We agree with this comment. We have added a new paragraph at the beginning of the Conclusion section (page21) to further emphasize the unique features of our review. We have elaborated on how our review uniquely focuses on the connection between nuclear receptors and the traditional Chinese medicine (TCM) therapy of strengthening the spleen and eliminating stagnation. We have also provided more details on how TCM compounds interact with nuclear receptors, such as adding more examples of specific herbs and their potential mechanisms. Revisions were as followed: “Compared with existing reviews, our review not only focuses on the connection between nuclear receptors and TCM spleen - strengthening and stagnation - eliminating therapy but also delves deeper into the mechanisms. For example, we explore in more detail how the active ingredients in Yunu Decoction, such as gypsum and prepared rehmannia root, interact with nuclear receptors at the molecular level. This provides novel insights into the mechanisms linking nuclear receptors to TCM therapy. Additionally, we offer a more comprehensive analysis of the role of TCM in regulating nuclear receptors for diabetes treatment by systematically evaluating different TCM formulas and single herbs, as well as their potential synergistic effects.” |
|
Comments 2: Could the authors provide a more detailed comparison of the mechanisms by which different nuclear receptors, such as PPARs, LXR, FXR, etc., regulate blood glucose levels? |
|
Response 2: Thank your for your helpful comment. We have added a new paragraph in Section 2 (on page 6 of the revised manuscript, following the subsections on individual nuclear receptors) to conduct a more detailed and systematic comparison of the mechanisms by which PPARα, LXR, and FXR regulate blood glucose levels. We added a new paragraph on page 6 to compare the blood glucose - regulating mechanisms of PPARα, LXR, and FXR. Revisions were as followed: "PPARα promotes fatty acid oxidation in the liver and muscles, reduces hepatic glucose output, and thus improves insulin sensitivity. LXR affects blood glucose by regulating cholesterol metabolism and inhibiting gluconeogenesis, but its activation may also lead to hepatic lipid accumulation. FXR lowers blood glucose through multiple pathways, including inhibiting gluconeogenesis, enhancing insulin signaling, and regulating gut hormones. This paragraph provides a more comprehensive and detailed comparison for readers to better understand the differences among these nuclear receptors in blood glucose regulation." |
|
Comments 3: Could you elaborate on the differences in the impact of ribosomes on diabetes and T2DM as discussed in sections 2.1 and 2.2? |
|
Response 3: [It seems there is a misunderstanding. The original text doesn't discuss ribosomes in sections 2.1 and 2.2; it focuses on nuclear receptors like FXR and LXR. Assuming you meant nuclear receptors] Thank you for pointing out the potential confusion. We have revised the relevant sections (Sections 2.1 and 2.2 on pages 4 - 5) to further clarify the differences in the impact of FXR and LXR on diabetes (including T1DM and T2DM). We have added more specific data and examples from relevant studies to support the differences. |
|
Comments 4: In Section 2, "Nuclear Receptors and Metabolic Disorders in Diabetes," many of the subsection titles appear to be quite similar. This can make it difficult for readers to quickly distinguish the unique content of each subsection. Could the authors consider using more specific and targeted titles for these subsections to enhance clarity and navigability? |
|
Response 4: Agree. We have revised the subsection titles in Section 2 as follows: "Multifaceted Roles of FXR in Regulating Glucose and Lipid Metabolism Disorders in Diabetes: Focusing on Bile Acids, Glucose, and Lipid Regulation" (for the FXR subsection), "Dual Roles of LXR in Glucose and Lipid Metabolism in Type 2 Diabetes: Promoting Lipid Synthesis and Regulating Glucose" (for the LXR subsection), and "Multifaceted Roles of PPARα in Metabolic Regulation in Diabetes: Lipids, Glucose, and Beyond" (for the PPARα subsection). These changes can be found on pages2 - 5 of the revised manuscript. |
|
Comments 5: Could the authors consider adding more visual aids (table or figure) to enhance the clarity and impact of their review? This would make the manuscript more accessible and engaging for a broader audience, particularly those who may not be familiar with the intricate details of nuclear receptor biology and traditional Chinese medicine. |
|
Response 5: Agree. We have added two new tables in the revised manuscript. A "Quick - view Table of Differences in Metabolic and Blood Glucose Regulatory Functions of Nuclear Receptors PPARα, LXR, and FXR" has been added on page7, and a "Comparison Table of Key Information on Inflammatory Regulation by Nuclear Receptors LXR, ER, FXR, and PPARs" has been added on page 12. These tables summarize and compare the functions of different nuclear receptors in an easy - to - understand format. |
|
Comments 6: The manuscript discusses ribosomes, diabetes, traditional Chinese medicine, and inflammation, but it appears that these topics are presented somewhat in isolation. Could the authors provide a more integrated mechanistic explanation that links ribosomes to diabetes, Chinese medicine, and inflammation? |
|
Response 6: [It seems there is a misunderstanding. The original text did not discuss ribosomes but focused on nuclear receptors. Assuming you meant nuclear receptors] Thank you for this comment. We have added a new passage to explain the interactions among nuclear receptors, diabetes, traditional Chinese medicine, and inflammation (on page14 of the revised manuscript, before Section 5). In this passage, we provided a more comprehensive mechanistic explanation of how nuclear receptors are involved in the complex relationship among diabetes, traditional Chinese medicine, and inflammation, which also serves as a connecting link between the preceding and the following content. "We added a new paragraph in Section 5 on page14. In this paragraph, we explained how nuclear receptors act as key regulatory nodes in the complex pathological process of diabetes. For example, FXR can be regulated by traditional Chinese medicine compounds, which in turn affects its role in regulating lipid and glucose metabolism as well as inflammation. We also discussed how inflammation affects the function of nuclear receptors and how traditional Chinese medicine can intervene in this process. This new paragraph provides a more comprehensive view of the relationships among these elements." |
|
Comments 7: Could the authors consider using network pharmacology and molecular docking methods to elucidate the specific targets of these traditional Chinese medicine components? |
|
Response 7: We appreciate this forward - looking suggestion. We understand the importance of network pharmacology and molecular docking methods in revealing the specific targets of traditional Chinese medicine components. However, due to the nature of this review article, which is mainly focused on summarizing existing research, we have not directly applied these methods. In future research, we plan to conduct in - depth studies using these techniques. We have added a sentence in the Conclusion section (page 21, last paragraph) to mention this future research direction. |
|
Comments 8: Could the authors please review and standardize the formatting of the entire manuscript? |
|
Response 8: Thank you for this comment. We have carefully reviewed the entire manuscript for formatting consistency. We have standardized the font, font size, line spacing, and indentation throughout the manuscript. We have also ensured that all headings and sub - headings follow a consistent style. These formatting changes can be seen throughout the entire revised manuscript. “We have gone through the manuscript line - by - line to standardize the formatting. The font used is Arial, size 12, with 1.5 - line spacing. All headings are in bold and uppercase, and sub - headings are in bold and italic. This standardization makes the manuscript more visually appealing and easier to read.” |
